# A Simple and Rapid Fungal DNA Isolation Assay Based on ZnO Nanoparticles for the Diagnosis of Invasive Aspergillosis

**DOI:** 10.3390/mi11050515

**Published:** 2020-05-19

**Authors:** Zhen Qiao, Huifang Liu, Geun Su Noh, Bonhan Koo, Qingshuang Zou, Kyusik Yun, Yoon Ok Jang, Sung-Han Kim, Yong Shin

**Affiliations:** 1Department of Convergence Medicine, Asan Medical Institute of Convergence Science and Technology (AMIST), University of Ulsan College of Medicine, 88 Olympicro-43gil, Songpa-gu, Seoul 05505, Korea; qiaozhen90@hotmail.com (Z.Q.); liuhuifang.1229@gmail.com (H.L.); ngs90@hanmail.net (G.S.N.); qhsgksdlek@naver.com (B.K.); zqs951108@gmail.com (Q.Z.); jangyo17@daum.net (Y.O.J.); 2Department of Bionanotechnology, Gachon University, Gyeonggi-do 13120, Korea; ykyusik@gachon.ac.kr; 3Department of Infectious Disease, Asan Medical Center, University of Ulsan College of Medicine, 88 Olympicro-43gil, Songpa-gu, Seoul 05505, Korea

**Keywords:** molecular diagnostics, medical device, clinical applications, invasive aspergillosis, zinc oxide nanoparticle

## Abstract

Invasive aspergillosis (IA) is an important cause of morbidity and mortality among immunocompromised people. Imaging and specimen tests used in the clinical diagnosis of aspergillosis with weak and indistinct defects leads to delay in the treatment of early aspergillosis patients. The developing molecular techniques provide a new method for the aspergillosis diagnosis. However, the existing methods are complex, time-consuming and may even be potentially hazardous. In this study, we developed a simple and rapid *Aspergillus fumigatus* spores DNA isolation assay using synthesized zinc oxide (ZnO). ZnO nanoparticles were used to take the place of the traditional commercial lysis buffer. The quality and quantity of the extracted DNA were sufficient for further diagnostics with polymerase chain reaction (PCR) analysis. This method offers easy, green, and economic alternative DNA isolation for the diagnosis of invasive aspergillosis.

## 1. Introduction

Invasive aspergillosis (IA) plays a significant role in the morbidity and mortality of immunocompromised individuals, bone marrow transplant recipients, cancer patients, human immuno-deficiency virus (HIV) patients, and patients undergoing treatment with immunomodulators [1,2]. There are more than 300 kinds of genus of *Aspergillus*, and *A. fumigatus* is the most prominent causative organism of IA. *A. fumigatus* is common in all environments but is difficult to distinguish from certain other molds (*A. flavus*, *A. niger*, and *A. terreus*) under the microscope [3,4]. Currently, the most common tests used to diagnose aspergillosis are the imaging tests using a chest X-ray or computerized tomography (CT) and specimen tests of sputum, tissue, and blood [5,6,7]. In respiratory secretion test, the patient’s sputum is stained with a dye and checked for the presence of *Aspergillus* filaments. The specimen is then placed in a culture that encourages the mold to grow to help confirm the diagnosis. Moreover, for the skin tissue test, *Aspergillus* antigen is injected into the patient’s forearm skin. A hard, red bump at the injection site appears if the patient has antibodies [8,9]. However, all these tests are time-consuming and uncomfortable, and additionally, due to the weak and indistinct clinical features of early IA, precious time is wasted in many patients requiring early diagnosis of the infection and prompt medical attention.

In the last two decades, developing molecular techniques have been widely used in the detection of fungal DNA in clinical samples. These molecular methods can identify clinical samples which usually contain a limited number of fungal cells with high sensitivity [10]. For the DNA samples used in the steps of downstream molecular analysis, besides the optimal primers and PCR conditions, it is exceedingly important to use the most efficient way to extract the DNA to detect low levels of *Aspergillus* DNA. Commonly, because of the fungal cell wall, isolating DNA from fungi is more difficult than from bacteria or mammalian cells and requires additional steps involving the use of enzymatic, mechanical, or chemical methods to disintegrate the fungal cell wall [11]. Currently, most commercial fungal DNA extraction kits used in clinics and laboratories are based on a standard method comprising the lyophilization of mycelia, disruption of the cell wall by grinding, extraction of DNA in a buffer containing sodium dodecyl sulfate, removal of proteins with a mixture of phenol and chloroform, and precipitation of DNA with 2-propanol [12,13,14]. Although this method can obtain large amounts of pure fungal DNA, it is time- and labor-intensive and presents with the potential threats of phenol and chloroform pollution. Therefore, a novel fungal DNA isolation assay would be desirable for simple, rapid, and highly sensitive detection of *Aspergillus* even at low levels.

Recent processes in nanomaterial synthesis and design have led to great advancements in the field of medicine, and nanomaterials also have their role in molecular diagnostic applications [15,16,17,18]. Metal nanoparticles, such as Ag, CuO, MgO, and ZnO, have attracted tremendous attention for their use in medical diagnostics [19,20,21,22]. ZnO is an emerging semiconductor which has a wide direct band gap (~3.3 eV) near the ultraviolet (UV) spectral region and large exciton binding energy (60 meV). Due to their unique chemical and physical properties, Zinc (Zn) and its oxide (ZnO) are widely applied in photoelectronics, sensors, electronics, photocatalysts, and antibacterial agents among other applications, and they have become the most interesting hotspot in this regard [23,24]. Actually, some promising studies have revealed that ZnO nanoparticles have an inhibitory effect on microbial growth and metabolism [25,26,27,28]. In our previous study, the nanomaterial ZnO was a good lysis buffer candidate for eukaryotic cells and common bacterial pathogens except fungi due to its limitation of not being able to break the cell membrane [29]. One paper has been published for fungal DNA extraction using zinc nanoparticle (ZnP) [30]. Although they have shown the utility of the ZnP for fungal DNA extraction, the method still requires alkaline lysis buffer and incubating temperature and an additional large instrument such as centrifugation. Thus, we are focusing upon developing a novel assay for fungal DNA isolation without any lysis buffer and large instruments including temperature controller. In this study, we first report an assay for fungal DNA isolation based on the effectiveness and performance of a cell lysis buffer at room temperature without additional instruments. ZnO-S-300 can disrupt *Aspergillus* cells by physical, chemical and biological methods [29]. In addition, the positive charge of ZnO-S-300 surface leads to bind the spore cell easily compared to the negative charge of commercial ZnOs surface. Therefore, the performance of this ZnO-based fungal DNA isolation assay is superior to those of other two kinds of commercial ZnO types, ZnO-C-100 and ZnO-C-5000.

## 2. Materials and Methods

### 2.1. Chemicals and Reagents

All reagents were of analytical grade and used without further purification. Zinc nitrate hexahydrate (Zn(NO_3_)_2_·6H_2_O, 98%), ammonium hydroxide solution (28% NH_3_ in H_2_O, 99.99% trace metals basis), and Tween20 (P2287) were obtained from Sigma-Aldrich (St. Louis, MO, USA). Hexadecyltrimethylammonium bromide (C_19_H_42_BrN >98%) was purchased from Tokyo Chemical Industry Co., Ltd (Tokyo, Japan). Commercial zinc oxide nanoparticles (dispersion, <100 nm, 20 wt% in H_2_O; Sigma-P Code: 721077) and commercial zinc oxide (powder, <5 μm, 99.9%; Sigma-P Code: 205532) were purchased as the substance for further characterization and comparison testing of the nanoparticles. Luria–Bertani medium was used for pathogen culture. Sabouraud dextrose agar with chloramphenicol media (Cat. No.C6781; Lot No.437412) was used for the A. fumigatus culture obtained from Santa Maria-USA. Milli-Q water with a resistance greater than 18 MΩ, 99% ethyl alcohol, phosphate-buffered saline (PBS, 10×, pH 7.4) were used in the experiments. 

### 2.2. Instruments and Kits

A commercial fungal DNA isolation kit, YeaStar Genomic DNA Kit^TM^ (Cat. No.D2002), was purchased from ZYMO RESEARCH (Irvine, USA. A C-Chip used as disposable hemocytometer was purchased from INCYTO (Cheonan, Republic of Korea). Field-emission scanning electron microscopy (FE-SEM) on a JSM-7500F instrument (JEOL, Seoul, Republic of Korea) was used to confirm the reaction and decoration materials. Zeta potentials of materials were measured using dynamic light scattering (DLS) on a DynaPro NanoStar instrument (Wyatt, Malvern, United Kingdom). Vortex mixer (T5AL) and a mini manufacturer (LABOGENE 1730R) MSH-30d stirring heater produced by Daihan Scientific Co., Ltd (Wonju-Si, South Korea) were obtained. Magic mixer (TMM-5) used for rotating the mixture tube during incubation was purchased from TOPSCIEN Instrument Co., Ltd (Ningbo, China). A CFX96 touch real-time polymerase chain reaction (PCR) detection system was purchased from BIO-RAD (Hercules, USA).

### 2.3. Fungal Samples

*Aspergillus fumigatus* (ATCC36607) was used in this study. It was grown in Sabouraud dextrose agar at 25 °C for 5 days. As the *A. fumigatus* grew, it produced spores; 1 mL 0.03% Tween 20 in water was used to collect the spores of *A. fumigatus*. After washing with phosphate-buffered saline (PBS), the *A. fumigatus* spore cells were counted using a hemocytometer.

### 2.4. Preparation of the ZnO Nanoparticles (ZnO-S-300) 

Herein, ZnO-S-300 nanoparticles were synthesized using the hydrothermal method in an alkaline medium [30]. For this, 0.1 M zinc nitrate hexahydrate (Zn(NO_3_)_2_·6H_2_O) and 0.1 M cetyltrimethylammonium bromide (CTAB) were mixed in 100 mL of sterilized deionized water (DI) in a 250-mL flask. The mixture was heated till 90 °C for 50 min with continuous stirring (500 rpm). The size of the ZnO that can be determined around 300 nm depends on the incubation time. Then, while maintaining the mixture at 90 °C with continuous stirring (500 rpm), 2 mL of ammonium hydroxide solution (NH_3_·H_2_O) was added at a rate of 1 mL/min, and during this period, the production of white precipitate was observed. Subsequently, the flask was transferred into an ice box and cooled down immediately to stop the reaction. The produced white precipitates were collected and washed with DI three times to wash away the residual ions. Then, we dried the white precipitates of ZnO at 56 °C in an oven.

### 2.5. *Aspergillus fumigatus* Spores’ Lysis with the ZnO-Based Fungal DNA Isolation Assay

The synthesized ZnO nanoparticles (ZnO-S-300) and commercial ZnO (ZnO-C-100 and ZnO-C-5000) were used for developing and optimizing the ZnO-based fungal DNA isolation assay. The workflow of the ZnO-based fungal DNA isolation assay is shown at the top in Figure 1. Herein, 100 μL of *A. fumigatus* spores was added into the 1.5-mL sample tube, and then, 100 μg (1 mg/mL, 100 μL) ZnO was added. Next, the mixture was put onto the rotator for 30 min rotation. After incubation, the mixture was centrifuged at 10,000 rpm for 1 min. Herein, the white ZnO was at the bottom of the tube. The supernatant was transferred into the Zymo-Spin III column and centrifuged at 10,000 rpm for 1 min. Then, 300 μL of DNA wash buffer was added and centrifuged for 1 min at 10,000 rpm to wash, and this wash step was repeated. Finally, the Zymo-Spin III column was transferred to a new 1.5-mL centrifuge tube, and 60 μL of water or TE was added directly onto the membrane. After waiting for 1 min, DNA was eluted by centrifugation at 10,000 rpm for 10 seconds.

### 2.6. *Aspergillus fumigatus* Spores’ Lysis with Commercial Kit

The commercial YeaStar Genomic DNA Kit^TM^ was used for comparing the cell lysis properties of the ZnO nanoparticles. The workflow for DNA extraction with the kit is shown at the bottom of Figure 1. For the first procedure, 10 μL *A. fumigatus* spores was added into the 1.5-mL sample tube, and 120 μL of YD digestion buffer and 5 μL of R-Zymolyase^TM^ (RNase A + Zymolyase^TM^) were added. These were mixed well by vortexing and incubated at 37 °C for 40–60 min. After incubation, 120 μL of YD lysis buffer was added into the mixture and gently vortexed. Next, 250 μL of chloroform was added and mixed thoroughly for 1 min, and the mixture turned from transparent to milky white. Then, the mixture was centrifuged in a table top centrifuge at 10,000 rpm for 2 min. At this point, the aqueous mixture had two layers, and the supernatant was transferred into the Zymo-Spin III column and centrifuged at 10,000 rpm for 1 min. Then, 300 μL of DNA wash buffer was added and centrifuged for 1 min at 10,000 rpm to wash, and this wash step was repeated. At last, the Zymo-Spin III column was transferred to a new 1.5-mL centrifuge tube and 60 μL of water or TE was directly added onto the membrane. After waiting for 1 min, DNA was eluted by centrifugation at 10,000 rpm for 10 seconds.

### 2.7. Quantitative Polymerase Chain Reaction (PCR) Condition

To analyze fungal DNA isolation performance, real-time PCR was performed after the isolation process. For the quantitative PCR process, the following procedure was implemented: 5 μL of DNA was amplified in a reaction mixture containing 10 μL of Green QPCR Master Mix (Agilent, #600882), 2.5 pmol of each primer, and 3 μL of DI water, and the total volume was made up to 20 μL. For detecting *A. fumigatus* DNA, we used the following primers: forward (5′-CACCCGTGTCTATCGTACCT-3′) and reverse (5′-ATTTCGCTGCGTTCTTCATC-3′). The PCR reactions were performed at 95 °C for 15 min, followed by 40 cycles of 95 °C for 10 s, 60 °C for 20 s, and 72 °C for 20 s, before a final elongation step at 95 °C for 10 s. 

## 3. Results and Discussion

### 3.1. Design of the ZnO-Based Fungal DNA Isolation Assay

Herein, we used the in-house synthesized ZnO nanomaterials to develop a fungal DNA isolation assay. The synthesized ZnO nanomaterials took the place of the traditional fungal cell lysis buffer and chloroform method to perform the fungal cell lysis process. This is briefly shown in Figure 1. Compared with the process of the commercial kit assay, this ZnO-based fungal DNA isolation assay showed several advantages. First, the whole DNA extraction process was fast and took approximately 45 min; this helps researchers not only save almost half the time but also lessen the workload. In commercial kit assays, after adding the digestion buffer, the mixture needed to be incubated at 37 °C for 40–60 min, followed by the addition of more lysis buffer and chloroform. Second, in the ZnO-based fungal DNA isolation assay, the use of ZnO instead of a common cell lysis buffer allows for fungal lysis within few minutes at room temperature without the need for an external thermal instrument. Third, it can also bypass the use of organic pollutants, such as chloroform. Generally, in most existing commercial kits, the input fungal cells are small volume (100-μL) of spores or the mycelia cell pellets after spinning down, and one of the reasons is the input sample solution can dilute the digestion buffer and lysis buffer; this approach has a limitation in that a sample with few cells cannot get into a pellet. On the other hand, the ZnO-based fungal DNA isolation assay is not limited by the input volume. 

### 3.2. ZnO Nanoparticles Effectively Work as Lysis Buffer

The synthesized ZnO nanoparticles (ZnO-S-300) which were compounded by the special formula that showed unique morphology (Figure 2A). The ZnO-S-300 was constituted by amounts of ZnO nanodots and had a polygonal shape. SEM images were used to confirm that ZnO-S-300 nanoparticles affected the fungal spores (Figure 2B) during the incubation process. Figure 2C shows that fungal spore cells were in physical contact with ZnO-S-300 nanoparticles. The membrane surface of untreated fungal spore cells was plump. After incubating with ZnO-S-300 nanoparticles (Figure 2C), these cells appeared crumpled. There are several hypotheses around the mechanism of the pathway by which ZnO-S-300 achieved cell split [30]. Regarding physical aspects, the specific shape and distinct positive zeta potential made it impossible for ZnO-S-300 to get into the cells. Regarding chemical aspects, the excessive Zn^2+^ released from the ZnO-S-300 was firmly adsorbed on the cell membrane surface by Coulomb’s law after reaching the membrane surface, and then, Zn^2+^ further penetrated the cell wall, causing cell wall rupture and the consequent cytoplasmic outflow [31,32]. Regarding biological aspects, the high-concentration ZnO-S-300 treatment triggered a reactive oxygen species (ROS) reaction in the tube environment, thus leading to apoptotic cell death and consequent cellular structure collapse [33,34]. In the case of the commercial kit, chaotropic reagents can disrupt the cell membrane of fungi [35]. As PCR cycle threshold (Ct) shown in Figure 2D, for the extraction by ZnO-based fungal DNA isolation assay, the cycle threshold (Ct) is 25.01 cycles, meanwhile, the cycle threshold (Ct) of the extraction by commercial kit assay is 22.92 cycles. The DNA product extracted by the commercially available kit and that extracted using the ZnO-based fungal DNA isolation assay had the same appointed melt temperature peak at 81.5 °C (Figure 2E). At the highest concentration of fungi (10^8^ cells), the performance of the kit was better than that of the ZnO-S-300 method due to lack of the amount of ZnO. 

### 3.3. Optimization of Synthesized ZnO Nanoparticles in the ZnO-Based Fungal DNA Isolation Assay

In foregoing tests, 100 μg (1 mg/mL, 100 μL) of synthesized ZnO nanoparticles was incubated with 10^8^. *A. fumigatus* spores in 100 μL of PBS for 60 min at room temperature. It was shown that this ZnO-based fungal DNA isolation assay could successfully extract the fungal DNA. Herein, we evaluated the DNA extraction efficiency of the ZnO-based fungal DNA isolation assay with different incubation temperatures and times using the quantitative PCR cycle threshold (Ct) as an indicator of DNA quality. Furthermore, we introduced two other commercial ZnO variants whose average sizes were ~100 nm (ZnO-C-100, Figure 3A) and ~5000 nm (ZnO-C-5000, Figure 3C). In the kit assay, 37 °C was the temperature used for activating and protecting R-Zymolyase. In the ZnO-based fungal DNA isolation assay, we explored the influence of temperature on the extraction process. Results as shown in Figure 3D revealed that ZnO-S-300 used at 55 °C and room temperature can both help procure the maximum quantity of DNA. The benefits of working at room temperature are that the heater step is eliminated and the word does not require large instruments. On comparing the three ZnO types, our synthesized ZnO-S-300 was the most effective in fungal DNA extraction. Furthermore, as effectiveness is another important index in a new operational approach, we checked the outcomes of the extraction by the three kinds of ZnO in different incubation time periods. As shown in Figure 3D, ZnO-C-5000 had few variations in extraction from 15 min to 60 min incubation periods, whereas ZnO-C-100 had good performance as lysis buffer at 30 min and 60 min incubation periods; remarkably, our synthesized ZnO-S-300 showed superior performance to the other two ZnOs at 30 min and 60 min incubation periods. In addition, for ZnO-S-300, the average Ct values at 30 min (27.07 cycles) and 60 min (26.35 cycles) incubation times were slightly different. The shortened incubation time improved work efficiency and reduced the burden of work. With regard to the performance of ZnO-S-300 being superior to that of ZnO-C-100 and ZnO-C-5000, one of the most crucial known reasons was that the synthesized ZnO-S-300 showed a special positive charge unlike the common ZnO, which has a negative charge; this property of ZnO-S-300 provided it with better affinity to negatively charged fungal spores and allowed for satisfactory lysis work.

### 3.4. Performance of ZnO-Based Fungal DNA Isolation Assay

The efficiency of both the ZnO-based fungal DNA isolation assay and commercial kit assays in different numbers of *A. fumigatus* spores is presented in Figure 4. In the CFX Maestro™ Software, the threshold of the quantitative PCR results was manually selected at 100 RFU (relative fluorescence units). Both methods can successfully extract DNA from PBS suspensions of *A. fumigatus* spores at a minimum of 10 cells. At higher densities of 10^8^ and 10^7^ spores, the commercial kit assay showed better performance than the ZnO-based fungal DNA isolation assay, whereas the latter persistently showed more sensitive and stronger signals than the former at densities of 10^6^ to 10^1^. The main cause of the ZnO-based fungal DNA isolation assay showing relatively inferior performance with densities of 10^8^ and 10^7^ is the deficient quantity of ZnO nanoparticles, whereas from the density of 10^6^, the proportion of ZnO and spores becomes optimal, and this ZnO-based fungal DNA isolation assay shows its advantages. High sensitivity of the ZnO-based fungal DNA isolation assay in low densities make it a good candidate for use with early invasive aspergillosis clinical samples which have few pathogens in samples. Although we have not shown the utility of this ZnO-based fungal DNA isolation assay in other fungal strains, we showed that the assay can isolate the DNA from bacteria (Figure A1).

## 4. Conclusions

In this study, a simple and rapid fungal DNA isolation assay based on ZnO nanoparticles for the diagnosis of invasive aspergillosis is presented. During the fungal DNA isolation process, the uniquely synthesized ZnO nanoparticles (ZnO-S-300) were used for the successful lysis of fungal spores. Meanwhile, this novel assay abates the chloride pollution and ponderous thermal instrument. With quantitative PCR analysis of the amplified DNA isolated using the ZnO-based fungal DNA isolation assay and commercial kit assay, our results show that these synthesized ZnO nanoparticles not only have physicochemical properties different from those of most commercial ZnO chemicals in the market but also perform better in the process of ZnO fungal DNA isolation than two kinds of commercial ZnOs. Our results also show that the ZnO-based fungal DNA isolation assay was more effective at detecting the fungi in samples with less density. Nevertheless, further study with clinical samples could be needed to validate this assay as a new DNA isolation assay from fungi. This study introduces a new candidate for diagnostic techniques for invasive aspergillosis.

## Figures and Tables

**Figure 1 micromachines-11-00515-f001:**
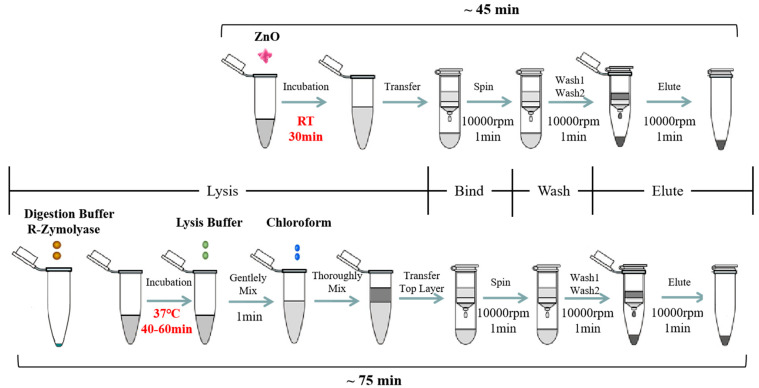
Diagram of the ZnO-based fungal DNA isolation assay. A schematic illustrating nucleic acid (NA) isolation using the lysis buffer replacement method based on ZnO (the ZnO-based fungal DNA isolation assay) instead of using the traditional lysis buffer in the commercial fungal DNA extraction kit. The ZnO-based fungal DNA isolation assay (**top**) and commercial kit assay (**bottom**) both have four steps involving different lysis buffers but the same filter column, wash buffer, and elution buffer. For lysis process, in the optimized ZnO-based fungal DNA isolation assay, 100 μL *A. fumigatus* spores was mixed with 100 μg ZnO for 30 min at room temperature only without any further lysis step. In the commercial kit assay, 10 μL *A. fumigatus spores* was mixed with 120 μL of YD digestion buffer and 5 μL of R-Zymolyase^TM^ (RNase A + Zymolyase^TM^) for 40–60 min at 37 °C, then YD lysis buffer and chloroform were added into the mixture. In the binding, washing, and elution steps, the buffer and columns used were the same in both methods. Subsequently, the NA extracted by both methods could be detected in downstream analysis.

**Figure 2 micromachines-11-00515-f002:**
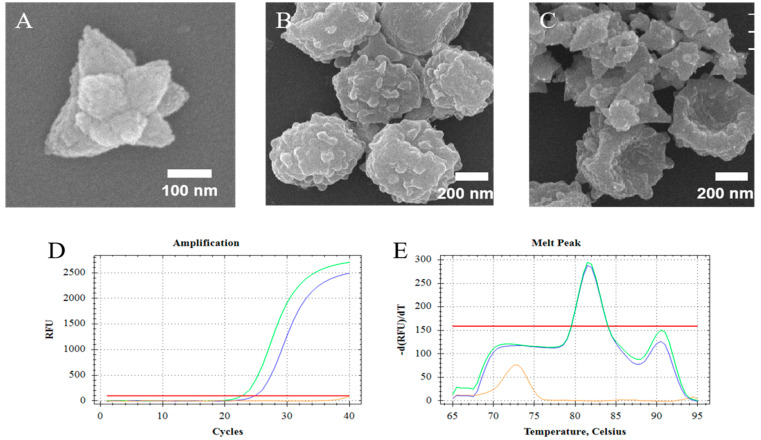
Study on the application of ZnO for the lysis of fungal spore cells. (**A**) Scanning electron microscope (SEM) images of synthesized nanoparticles ZnO-S-300; (**B**) SEM images of untreated *A. fumigatus* spore cells; (**C**) SEM image of the lysis process of ZnO-S-300 on *A. fumigatus* spore cells. (**D**,**E**) Fluorescence signals and melt data from real-time PCR analyses of amplified DNAs extracted by the ZnO-based fungal DNA isolation assay (Blue line) and kit assay (Green line). For the ZnO-based fungal DNA isolation assay, the cells were incubated for 60 min at room temperature. The extracted DNA of *A. fumigatus* spore cells (10^8^ cells) was eluted in 100-μL elution buffer. The orange line was the negative control which is pure elution buffer. Fluorescence intensity is measured in relative fluorescence units (RFU) by real-time polymerase chain reaction (PCR).

**Figure 3 micromachines-11-00515-f003:**
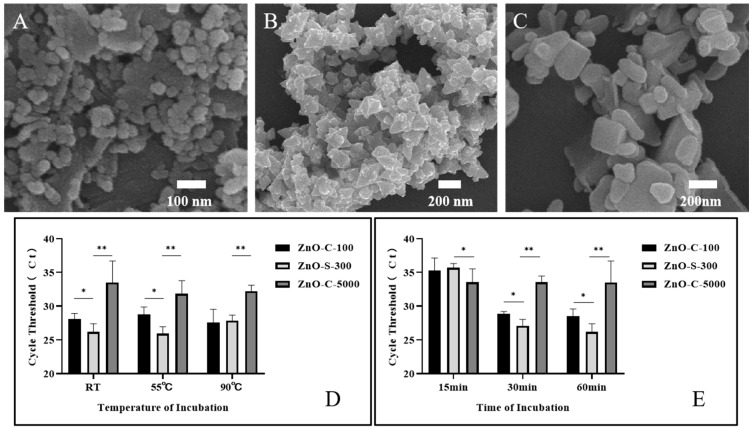
Optimization of the work conditions for ZnO nanoparticle synthesis in the ZnO extraction method. (**A**) SEM images of commercial nanoparticles ZnO-C-100. (**B**) SEM images of synthesized nanoparticles ZnO-S-300, (**C**) SEM images of commercial nanoparticles ZnO-C-5000. (**D**,**E**) Performance evaluation of the ZnO lysis buffer DNA extraction method using the synthesized ZnO (~300 nm, ZnO-S-300) nanomaterial at three different incubation temperatures (**D**) and times (**E**) by comparing the cycle threshold (Ct) of real-time PCR against two other commercially available ZnO types (~100 nm,~5000 nm). The extracted DNA of *A. fumigatus* spore cells (10^8^ cells) was eluted in 100-μL elution buffer. Error bars indicate standard deviation from the mean based on at least three independent experiments. * *p* < 0.05, ** *p* < 0.01.

**Figure 4 micromachines-11-00515-f004:**
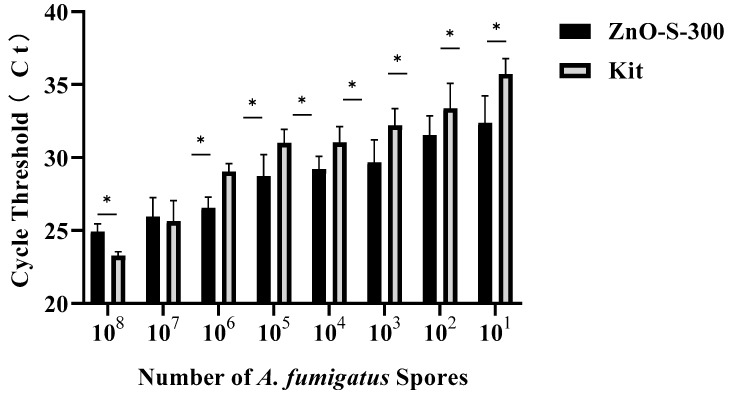
Performance evaluation of the DNA extraction assay using either the ZnO-based fungal DNA isolation assay or the commercial kit assay with *A. fumigatus* spore concentrations ranging from 1 to 10^8^ cells/100 μL by comparing the cycle threshold (Ct) of real-time PCR. Error bars indicate standard deviation from the mean, based on at least three independent experiments. Error bars indicate standard deviation from the mean, based on at least three independent experiments. * *p* < 0.05.

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
