# Peer review of "A Simple and Rapid Fungal DNA Isolation Assay Based on ZnO Nanoparticles for the Diagnosis of Invasive Aspergillosis"

_micromachines, 2020, doi:10.3390/mi11050515_

Round 1

Reviewer 1 Report

The Manuscript represents a very interesting and useful topic on how to simplify and decrease the time needed for DNA extraction using ZnO nanoparticles that can overcome the performances of the commercially available solutions.

Not being a novel approach (other ZnO-based nanoparticles exist), this manuscript presents a conceptually noteworthy idea to simplify the present detection technologies. Nevertheless, some additional experiments are needed to have more scientific soundness and to prove the effectiveness of the proposed method.

The main concerns and comments:

  1. The documented procedure is not an “alternative for the diagnosis of invasive aspergillosis”. Instead it is an alternative method for DNA isolation that will complement the current detection techniques (PCR) and contribute to their simplification (Abstract section).
  2. The study was performed using only one reference A. fumigatus strain. But morphological differences appear when cells are exposed to different biological effects and between different strains. Saying that, I firmly believe that it should be considered the testing of other strains, namely isolates from different origins. In addition other Aspergillus species should also be considered. This step is not conferring specificity to the test, and other species (yet in less extent) are also able to cause IA.
  3. The performance of the protocol should be proved with real samples (BAL, Sputum,…). The type and condition of clinical sample can influence the results.
  4. Line 186: it is refer that “ZnO-based fungal DNA isolation assay is useful for all types of fungal samples, such as spores and mycelia cells, with no input volume or medium”. This has to be substantiated by literature. In addition no experiments in this manuscript support this idea, since only spores were tested.

Other comments:

  1. In introduction is missing information on how ZnO is able to disrupt Aspergillus cells and the difference on the proposed and commercially available ZnO nanoparticles.
  2. 4. Preparation of the ZnO nanoparticles: I can’t judge the correctitude of the proposed preparation protocol, because I don’t have the necessary background, but I am curious on how can be guaranteed the 300nm size. Maybe a brief explanation on that will elucidate better on the specific synthesis.
  3. Spores lysis using ZnO-based assay and commercial kit (YeaStar Genomic DNA Kit): Why the authors started with a different fungal spores volume (100ul for ZnO-based assay and only 10 ul for the commercial kit)? This should not influence the results? What would happen if the ZnO-based assay started with smaller volumes? How the authors choose to used these particular volumes?
  4. Figure 1 – In here the authors are assuming the assessed conditions (RT, 30 min). However, the conditions optimization results were only presented further in the manuscript. Ideally, to be more rigorous, I rather take off the specific conditions on Figure 1. Otherwise, it is possible to refer on the subtitles that conditions were optimized on the evaluation assays.
  5. Figure 2 – Explain what is RFU in subtitles.
  6. 2. ZnO nanoparticles effectively work as lysis buffer:
    • Line 207-214: Please give a more in deep explanation for these results. As it is a “results and discussion” section, simple assumptions without any support from available literature it’s not advisable. What happen to cells when applied the commercial kit? It should be interesting to have some morphological information for that as well.
    • Line 213: please give the full term for ROS
    • Line215: What implications these results have to ZnO-based DNA extraction? The Ct values are higher using ZnO.
  1. 3.3. The optimization assays aimed to evaluate the best conditions of ZnO-S-300, comparing with other ZnO-based nanoparticles. I assume the authors used that optimized conditions to further assess the ZnO nanoparticles effectiveness as lysis buffer. Hence, I advise the authors to consider reordering the manuscript in order to give correct timetable information and be more rigorous (see above).

Author Response

We thank the reviewers for their thoughtful review of our manuscript. We have taken their comments into careful consideration in preparing our revised manuscript, significantly improving its quality. Below, we present our responses to the individual comments. The detailed responses, please see the attachment.

Reviewer #1:

The Manuscript represents a very interesting and useful topic on how to simplify and decrease the time needed for DNA extraction using ZnO nanoparticles that can overcome the performances of the commercially available solutions.

Not being a novel approach (other ZnO-based nanoparticles exist), this manuscript presents a conceptually noteworthy idea to simplify the present detection technologies. Nevertheless, some additional experiments are needed to have more scientific soundness and to prove the effectiveness of the proposed method.

The main concerns and comments:

  1. The documented procedure is not an “alternative for the diagnosis of invasive aspergillosis”. Instead it is an alternative method for DNA isolation that will complement the current detection techniques (PCR) and contribute to their simplification (Abstract section).
  • We thank the reviewer for the comment.According tothe comments, we have corrected it on Abstract Section in the revised manuscript.
  • On page 1 (Abstract section), “This method offers an easy, green, and economic alternative DNA isolation for the diagnosis of invasive aspergillosis.”

  1. The study was performed using only one reference A. fumigatus strain. But morphological differences appear when cells are exposed to different biological effects and between different strains. Saying that, I firmly believe that it should be considered the testing of other strains, namely isolates from different origins. In addition other Aspergillus species should also be considered. This step is not conferring specificity to the test, and other species (yet in less extent) are also able to cause IA.
  • We thank the reviewer for the comment. Due to the COVID-19 pandemic, there is some problem for doing additional experiment using other biological samples. We are really sorry for this situation. Thus, we have examined the utility of the ZnO-S assay for extraction of DNA from other bacteria (Gram-Negative and Gram-Positive). We have added the result on page 8 and the supplementary file of the revised manuscript.
  • On Page 8, “Although we have not shown the utility of this ZnO-based fungal DNA isolation assay in other fungal strains, we showed that the assay can be isolated DNA from other bacteria (Fig. S1).”

FigureS1. Synthesized ZnO nanoparticles for lysis of bacterial cells. We used 100μL (104 CFU/mL)other bacterial species including Gram-positives (Staphylococcus aureus and Bacillus cereus) and Gram-negatives (Escherichia coli and Brucella ovis) to test of the performance of theZnO-based DNA isolation assay. * p < 0.05

  1. The performance of the protocol should be proved with real samples (BAL, Sputum,…). The type and condition of clinical sample can influence the results.
  • We thank the reviewer for the comment. Due to the COVID-19 pandemic, there is some problem for doing additional experiment with real samples in 10 days. We are really sorry for this situation. Nevertheless, further study with clinical samples could be needed to validate this assay as a new DNA isolation assay from fungi.We have added it on page 8 in the revised manuscript.
  • On page 8, “Nevertheless, further study with clinical samples could be needed to validate this assay as a new DNA isolation assay from fungi.”

  1. Line 186: it is refer that “ZnO-based fungal DNA isolation assay is useful for all types of fungal samples, such as spores and mycelia cells, with no input volume or medium”. This has to be substantiated by literature. In addition no experiments in this manuscript support this idea, since only spores were tested.
  • We thank the reviewer for the comment. According to the comments, we have deleted the sentence on page 5 in the revised manuscript.

Other comments:

  1. In introduction is missing information on how ZnO is able to disrupt Aspergillus cells and the difference on the proposed and commercially available ZnO nanoparticles.
  • We thank the reviewer for the comment. According to the comments, we have added the information on page 2 in the revised manuscript.
  • On Page 2, “ZnO-S-300 can disrupt Aspergillus cells by physical, chemical and biological ways [30]. In addition, the positive charge of ZnO-S-300 surface leads to bind the spore cell easily compared to the negative charge of commercial ZnOs surface. Therefore, the performance of this ZnO-based fungal DNA isolation assay is superior to those of other two kinds of commercial ZnO types, ZnO-C-100 and ZnO-C-5000.”

  1. 4. Preparation of the ZnO nanoparticles: I can’t judge the correctitude of the proposed preparation protocol, because I don’t have the necessary background, but I am curious on how can be guaranteed the 300nm size. Maybe a brief explanation on that will elucidate better on the specific synthesis.
  • We thank the reviewer for the comment. For the synthesis of the ZnO-S-300, we used the method previously reported [30]. Briefly, we mixedthe precursor Zn(NO3)2·6H2O and the template CTAB in the hot waterandincubate for 50min. Then,NH4OH was added as precipitating agent to get the white homogeneous ZnO precipitating.Finally, we checked the size of ZnO around 300nm. We have added the information on page 3 in the revised manuscript.
  • On Page 3, “Herein, ZnO-S-300 nanoparticles were synthesized using the hydrothermal method in an alkaline medium [30]. For this, 0.1 M zinc nitrate hexahydrate (Zn(NO3)26H2O) and 0.1 M cetyltrimethylammonium bromide (CTAB) were mixed in 100 mL of sterilized deionized water (DI) in a 250-mL flask. The mixture was heated till 90°C for 50 min with continuous stirring (500 rpm). The size of the ZnO can be determined around 300 nm depends on the incubation time.”

  1. Spores lysis using ZnO-based assay and commercial kit (YeaStar Genomic DNA Kit): Why the authors started with a different fungal spores volume (100ul for ZnO-based assay and only 10 ul for the commercial kit)? This should not influence the results? What would happen if the ZnO-based assay started with smaller volumes? How the authors choose to used these particular volumes?
  • We thank the reviewer for the comment. Based on thecommercial kit’s protocol, the input samplevolumeshould be around 10 ul after the centrifugation. On the other hand, the ZnO-S-300 method can be used at various sample volume (10 ul to 1 ml) for the lysis step.
  • Nevertheless, we used same sample concentrations (10 to 108cells) of fungi in 10 ul and 100 ul volume, respectively. Therefore, the volume does not influence the results. We have added the information on page 5 in the revised manuscript.
  • On Page, “On the other hand, the ZnO-based fungal DNA isolation assay is not limited by the input volume.”

  1. Figure 1 – In here the authors are assuming the assessed conditions (RT, 30 min). However, the conditions optimization results were only presented further in the manuscript. Ideally, to be more rigorous, I rather take off the specific conditions on Figure 1. Otherwise, it is possible to refer on the subtitles that conditions were optimized on the evaluation assays.
  • We thank the reviewer for the comment. According to the comments, we have corrected the figure 1 on page 4 in the revised manuscript.
  • On Page 4, “Figure 1.Diagram of theZnO-based fungal DNA isolation assay. A schematic illustrating nucleic acid (NA) isolation using the lysis buffer replacement method based on ZnO (the ZnO-based fungal DNA isolation assay) instead of using the traditional lysis buffer in the commercial fungal DNA extraction kit. The ZnO-based fungal DNA isolation assay (top) and commercial kit assay (bottom) both have four steps involving different lysis buffers but the same filter column, wash buffer, and elution buffer. For lysis process, in theoptimizedZnO-based fungal DNA isolation assay, 100 μL fumigatusspores was mixed with 100 μg ZnO for 30 min at room temperatureonly without any further lysis step. Whereas in the commercial kit assay, 10 μL A. fumigatusspores was mixed with 120 μL of YD Digestion Buffer and 5 μL ofR-ZymolyaseTM(RNase A + ZymolyaseTM) for 40–60 min at 37°C, then YD lysis buffer and chloroform were added into the mixture. In the binding, washing, and elution steps, the buffer and columns used were the same in both methods. Subsequently, the NA extracted by both methods could be detected in downstream analysis.”

  1. Figure 2 – Explain what is RFU in subtitles.
  • We thank the reviewer for the comment.According to the comments, we have added the information on page 5 in the revised manuscript.
  • On Page 5, Figure 2 legend, “Fluorescence intensity is measured in relative fluorescence units (RFU) by real-time PCR.”

  1. 2. ZnO nanoparticles effectively work as lysis buffer:
    • Line 207-214: Please give a more in deep explanation for these results. As it is a “results and discussion” section, simple assumptions without any support from available literature it’s not advisable. What happen to cells when applied the commercial kit? It should be interesting to have some morphological information for that as well.
  • We thank the reviewer for the comment. According to the comments, we have added more references [32-35] in the revised manuscript.

    • Line 213: please give the full term for ROS
  • We thank the reviewer for the comment. According to the comments, we have added the information (reactive oxygen species =>ROS) in the revised manuscript.

    • Line215: What implications these results have to ZnO-based DNA extraction? The Ct values are higher using ZnO.
  • We thank the reviewer for the comment. According to the comments, we have added the information in the revised manuscript.
  • On Page 6, “At highest concentration of fungi (108cells), the performance of the kit was better than that of the ZnO-S-300 method due to lack of the amount of ZnO.”

  1. 3. The optimization assays aimed to evaluate the best conditions of ZnO-S-300, comparing with other ZnO-based nanoparticles. I assume the authors used that optimized conditions to further assess the ZnO nanoparticles effectiveness as lysis buffer. Hence, I advise the authors to consider reordering the manuscript in order to give correct timetable information and be more rigorous (see above).
  • We thank the reviewer for the comment.According to the comments, we have modified Figure1 in the revised manuscript.

Figure 1.Diagram of the ZnO-based fungal DNA isolation assay

Reviewer 2 Report

The paper demonstrated a simple and rapid fungal DNA isolation assay based on ZnO nanoparticles for the diagnosis of invasive aspergillosis is presented. They used a uniquely synthesized ZnO nanoparticle (ZnO-S-300) for the lysis of fungal spores. Their results showed that the ZnO-based fungal DNA isolation assay was more effective to detect the fungi in samples with less density. Several issues should be clarified before the further consideration of publication.

  1. Spelling typos and wrong grammars throughout the article, listed several as followed, please check and revise.

Line 25  wrong keyword “Apergillosis”.

Line 165 “Where as” should be whereas.

Line 182-183 “extra thermal instrument” wrong grammar expression.

Line 237 “ others” should be other

Line 251 “difference” should be different

Line 275 “make” should be makes

  1. Authors claimed that their method is more rapid than commercial kit assays, can authors express this improvement in a timeline table? This can help readers and researchers better understand your contribution.

  1. According to the previous report “ Al-Dhabaan, Fahad A., et al. "Enhancement of fungal DNA templates and PCR amplification yield by three types of nanoparticles." Journal of Plant Protection Research58.1 (2018).”, can authors explain the novelty of your work and also compare the results with this reference?

Author Response

We thank the reviewers for their thoughtful review of our manuscript. We have taken their comments into careful consideration in preparing our revised manuscript, significantly improving its quality. Below, we present our responses to the individual comments. The detailed responses, please see the attachment.

Reviewer #2:

The paper demonstrated a simple and rapid fungal DNA isolation assay based on ZnO nanoparticles for the diagnosis of invasive aspergillosis is presented. They used a uniquely synthesized ZnO nanoparticle (ZnO-S-300) for the lysis of fungal spores. Their results showed that the ZnO-based fungal DNA isolation assay was more effective to detect the fungi in samples with less density. Several issues should be clarified before the further consideration of publication.

  1. Spelling typos and wrong grammars throughout the article, listed several as followed, please check and revise.

 Line 25  wrong keyword “Apergillosis”.

Line 165 “Where as” should be whereas.

Line 182-183 “extra thermal instrument” wrong grammar expression.

Line 237 “ others” should be other

Line 251 “difference” should be different

Line 275 “make” should be makes

  • We thank the reviewer for the comment.According to the comments, we have corrected these in the revised manuscript.

  1. Authors claimed that their method is more rapid than commercial kit assays, can authors express this improvement in a timeline table? This can help readers and researchers better understand your contribution.
  • We thank the reviewer for the comment.According to the comments, we have modified Figure1 in the revised manuscript.

  1. According to the previous report “Al-Dhabaan, Fahad A., et al. "Enhancement of fungal DNA templates and PCR amplification yield by three types of nanoparticles." Journal of Plant Protection Research58.1 (2018).”, can authors explain the novelty of your work and also compare the results with this reference?

  • We thank the reviewer for the comment. After carefully reading the suggested paper, we have added the reference paper in introduction part of the revised manuscript.

On Page 2, “One paper has been published for fungal DNA extraction using zinc nanoparticle (ZnP) [31]. Although they have showed the utility of the ZnP for fungal DNA extraction, the method still required alkaline lysis buffer and incubating temperature and additional large instrument such as centrifugation.

Round 2

Reviewer 1 Report

In view of the comments sent, the authors were able to explain
clearly and make modifications to the manuscript that allow
the article to be more clear.
Line 190 It should be written "external"

Reviewer 2 Report

The manuscript now is accepted for further consideration of publication.